# Musculoskeletal and Gait Characteristics in Patients with Stickler Syndrome: A Cross-Sectional Study

**DOI:** 10.3390/children9121895

**Published:** 2022-12-02

**Authors:** Juan José Fernández-Pérez, Paloma Mascaraque-Ruiz, Carlos Martín-Gómez, Ignacio Martínez-Caballero, Teresa Otón, Loreto Carmona, Sergio Lerma-Lara

**Affiliations:** 1Toledo Physiotherapy Research Group (GIFTO), Faculty of Physiotherapy and Nursing, Castilla la Mancha University, 45004 Toledo, Spain; 2Facultad de Ciencias de la Salud, CSEU La Salle, UAM, 28023 Madrid, Spain; 3Fundación Hospital Universitario Niño Jesús, 28009 Madrid, Spain; 4Hospital Infantil Universitario Niño Jesús, 28009 Madrid, Spain; 5Instituto de Salud Musculoesquelética, 28045 Madrid, Spain

**Keywords:** Stickler syndrome, collagenopathy, 3D gait analysis, quantitative ultrasound stiffness index

## Abstract

Background: Stickler syndrome (SS) is a connective tissue disorder of fibrillary collagen with very variable clinical manifestations, including premature osteoarthritis and osteopenia. This musculoskeletal alteration may affect gait maturity or produce strength difficulties. Objective: Our aim was to describe the musculoskeletal characteristics, bone stiffness, gait kinematics, and kinetics of SS patients. Methods: This is a cross-sectional study of children and youngsters with SS recruited by telephone calls through the Spanish SS Association. All participants underwent an analysis of musculoskeletal characteristics, including a 3D gait analysis. Results: The sample included 26 SS patients, mainly boys (65.4%) with a median age of 11 (IQR 5–14). The manual muscle testing was normal in 88.5% of patients. The median distance covered in the 6-min walking test was 560.1 ± 113.4 m. Bone stiffness index scores were 70.9 ± 19.7 for children under 10 years and 88.3 ± 17.5 for children older than 10 years. The gait indicators GPS and GDI were: 7.4 ± 1.9 and 95.3 ± 9.7, respectively, for the left side and 6.8 ± 2.0 and 97.7 ± 9.5 for the right side, respectively. Conclusions: In our series of patients with SS, we found muscle-articular involvement does not have a high impact on strength or gait problems. More work is needed to understand the effect of SS on the musculoskeletal system.

## 1. Introduction

Stickler syndrome (SS) was described in 1965 by Stickler et al. [1]. It is a dominantly inherited connective tissue disorder of fibrillary collagen with high variability in the manifestation of phenotypes [2,3]. It has an estimated incidence of 1 case per 10,000 births. Collagen is an extracellular fibrous protein that forms part of the connective tissue and is especially abundant in weight-bearing tissues such as cartilage, bone, tendons, fascia, and dermis. It is also the framework for all organs and tissues. There are 40 different genes that encode at least 27 different types of collagen [4].

SS is produced by heterogeneous mutation in four genes that control the synthesis of collagen 2, 9, and 11, so it has a very variable phenotypic expression. The responsible mutations are in COL2A1, COL11A1, COL11A2, and COL9A1 procollagen genes, leading to various degrees of abnormal synthesis collagen types II, XI, or IX [5]. Collagen 2 is found in the greatest proportion in the vitreous humor, cartilage and intervertebral discs. Collagen 9 is associated with type 2 collagen fibrils in mature articular cartilage, the cornea, and vitreous humor. Collagen 11 has a distribution similar to that of type 2. The three types of collagen are found in the cochlea. It is characterized by congenital conditions (i.e., megalophthalmos, retinal detachment, deafness, cleft palate, Pierre Robin sequence, joint hypermobility, and premature arthritis) [4]. 

SS is classified into different types according to the mutated gene, which explains the ophthalmological phenotype and, specifically, the anomalies in the architecture of the vitreous. Based on the vitreous abnormalities, Stickler syndrome is classified as type 1 (“membranous”, which is characterized by a persistence of vestigial vitreous gel in the retrolental space) and type 2 (“beaded”, which is characterized by sparse and irregularly thickened bundles throughout the vitreous cavity) [6,7].

The clinical manifestations are very variable, generally distributed in four large groups: (A) craniofacial findings: may include a flat facial profile, telecanthus and epicanthal folds, micrognathia, and cleft palate; (B) eye alterations: early cataracts and nonprogressive myopia are common; (C) hearing impairment, especially sensorineural deafness for high tones, is common but overall sensorineural hearing loss in type I Stickler syndrome is typically mild and not significantly progressive [8,9]; and (D) musculoskeletal features, specifically early onset arthropathy, short stature, and mild spondyloepiphyseal dysplasia. In children and adolescents, joint hypermobility is seen and usually becomes less prominent with age. Other manifestations in these patients are skeletal alterations related to orthopedic problems. They also experience frequent spinal abnormalities such as scoliosis, Scheuermann-like kyphosis deformities and spondylolisthesis [10,11]. Premature osteoarthritis and osteopenia are also frequent in these patients. There also appears to be a predisposition to femoral head complications such as Legg–Perthes disease or slipped epiphysis [3,11,12].

The presence of musculoskeletal disorders in children may affect gait maturity or may result in strength difficulties. The aim of this study was to analyze the musculoskeletal characteristics, gait kinematics, and kinetics of SS patients. 

## 2. Materials and Methods

A cross-sectional study was conducted. The study protocol and materials were approved by the ethics committee of the Centro Superior de Estudios Universitarios LaSalle (CSEULS) in Madrid, Spain. The outcomes measured were taken on two days: one for musculoskeletal characteristics and the other for gait analysis and a walking test.

Patients were eligible for the study if they had a medical diagnosis of SS, their age was between 4 and 18 years, and they had the ability to walk at least eight meters. Eligible participants were recruited from December 2017 to March 2018 by telephone call via the Spanish Stickler Syndrome Association roll. 

We recruited 26 participants for this study. Patients and their family were received into La Salle M-Lab, signed the informed consent, and were interviewed by one of the researchers. Information was collected on the following three groups of variables: 

### 2.1. Musculoskeletal Characteristics, Muscular Strength, and Functional Tests

Information on the clinical characteristics of the musculoskeletal system (osteoarthritis, joint hiperlaxitude, marfanoid habit, spinal dysplasia, muscular atrophy, etc.) was collected. Muscular strength was evaluated with manual muscle testing (MMT) following the Medical Research Council muscle strength scoring system [13]. MMT is used in rehabilitation and recovery to evaluate contractile units, including muscles and tendons, and their ability to generate forces (score range 0–5; minimum 0, maximum 5/5). A score of 3 or higher is considered normal. In addition, the following functional tests were performed: (a) In the Duncan–Ely test (assessment of rectus femoris spasticity or tightness), the patient lies prone in a relaxed state. The test is positive when the heel cannot touch the gluteus maximus or the hip of the tested side rises from the table [14]. (b) The Galeazzi test or Allis’ sign (exploration of hip dislocation or dysplasia) is performed by flexing the infant’s knees when they are lying down so that the feet touch the surface and the ankles touch the buttocks. If the knees are not level, the test is positive [15]. (c) Thomas test (assessment of the flexibility of the hip flexors). A test is positive (if the iliopsoas muscle is shortened, or a contracture is present) when the lower extremity on the involved side is unable to fully extend at the hip [16]. (d) The Silfverskiöld test differentiates gastrocnemius tightness from an Achilles tendon contracture by evaluating ankle dorsiflexion with the knee extended and then flexed [17]. We counted how many of the included patients presented a pathological result on the different tests.

### 2.2. Calcaneus Quantitative Ultrasound (QUS)

QUS is a quick, cost-efficient, and radiation-free method used to evaluate bone stiffness and indicates the density, structure, and composition of the bone [18,19]. Specifically, the calcaneus was found to be a reliable location to assess bone status. The calcaneus consists of 90% trabecular bone, which shows a high metabolic rate. The bone microarchitecture is similar to that of the lumbar spine and femoral neck, which are major body sites for diagnosing osteoporosis [18]. The results of the QUS of calcaneus are expressed as the stiffness index (SI), a composite of speed of sound (SOS) and broadband ultrasound attenuation (BUA). The stiffness index (SI) is calculated from the BUA and SOS in the Achilles system according to the following equation: SI = [(0.67 × BUA + 0.28 × SOS) − 420] [20]. Previously, QUS calcaneal SI showed moderate correlation (r = 0.69) with total body bone mineral density measured by dual-energy X-ray absorptiometry [21]. QUS was measured using an Achilles EXP II system (Getz Healthcare, Bangkok, Thailand).

### 2.3. Gait and Walking Test

For 3D gait analysis, three-dimensional gait analysis was used for obtaining information on the kinetic (power) and kinematic (joint motion) parameters in the three planes: sagittal, frontal, and transverse. Eight optoelectronic cameras (BTS BioEngeneering SMART-DX 6000) and two BTS P-6000 dynamometric platforms were used to collect kinematic data, sampling at 250 Hz, and the components of forces in the three coordinate axes. The modified Helen–Hayes marker set was used to quantify the kinetics and kinematics of the lower limbs joints [22]. Patients were instructed to: (1) stand on the platform in anatomical position; (2) walk barefoot along an 8-m walkway, with 3 trials collected. Image capture (Smart-Capture BTS BioEngeneering) and Visual3D (Smart-Clinic BTS BioEngeneering) programs were used to track, process, and compare the results with normal values of the kinetics and kinematic data. To confirm the absence or presence of gait pathology, we used the gait profile score (GPS) and gait deviation index (GDI) [23,24]. The GDI is a global measure that provides a numerical value that expresses gait pathology (ranging from 0 to 100, where 100 indicates the absence of gait pathology). The GDI is more complete, less ambiguous, show better statistical performance, and is easier to use than the Gillette gait index [23,25]. In addition, the data obtained during 3D gait analysis were compared with those from a sample of healthy children and adolescents (n = 25) from the Hospital Infantil Universitario Niño Jesús, with the aim of describing the alterations in gait by comparing them with reference values in children and adolescents of similar ages. 

A 6-min walking test (6MWT) was also performed. It is a submaximal exercise test that entails measurement of distance walked over a period of 6 min. 

### 2.4. Statistical Analysis

The statistical analysis was carried out using IBM Corp. (Released 2020. IBM SPSS Statistics for Windows, Version 27.0. Armonk, NY, USA: IBM Corp.) Non-parametric tests were used due to the small sample size. Summary measures (median and interquartile range (IQR) for sample characteristics; 95% confidence interval (CI) for results obtained in quantitative measures and percentages for dichotomous outcomes) are used to describe the sample. Possible differences between the groups were explored by the Mann–Whitney U and chi-square tests for quantitative and qualitative variables, respectively. No imputation was performed for missing data.

## 3. Results

The sample consisted of 26 subjects, of whom 9 (34.6%) were girls, with a median age of 11.0 years (IQR 5–14). The median weight was 40.8 kg (IQR 21.4–55.9) and the median height was 1.5 m (IQR 1.2–1.6). The MMT (muscle examination) showed that almost all patients had normal strength. Of all the muscle groups explored, 88.5% of patients had a value of three or more points (which could be considered normal or almost normal). Only two patients (7.7%) had a muscle group with altered strength, and one patient (3.8%) had two muscle groups with a score of less than three over five in manual muscle testing. 

The sample lost was n = 3 for musculoskeletal features, n = 1 for functional test, and n = 4 for 6MWT due to missing data and the fact that some children refused to participate in these measurements.

### 3.1. Musculoskeletal Characteristics and Physical Examination

Table 1 presents a summary of the main musculoskeletal features of these patients with SS. The most common alteration was ligament hyperlaxity syndrome (30%), followed by osteoarthritis, marfanoid habit, and muscular atrophy (17% each one). In the section of functional tests, the number and percentage of patients with altered tests are shown. 

Twenty-two patients completed the 6MWT. The mean distance covered was 560.1 m (±113.4), with minimum and maximum values of 360 and 729 m, receptively (the median was 575, IQR 458–653). Table 2 shows the results by sex (without significant differences) and by age (the distance travelled by the oldest children being significantly greater; *p* value = 0.02).

### 3.2. Calcaneus Quantitative Ultrasound, QUS

QUS was performed in 26 patients. The QUS results were as follows: the median values were 1555.3 (IQR 1545.5–1606.5) for SOS value, 91.9 (IQR 74.8–106.2) for BUA, and 77.5 (IQR 61–99) for SI. Likewise, a subanalysis was carried out by sex (no differences were found) and by age (with significant differences between those younger and older than 10 years; *p* value = 0.02) (Table 2). Previously, the QUS SI showed a moderate correlation (r = 0.69) with total body bone mineral density measured by dual-energy X-ray absorptiometry [21].

### 3.3. Gait Analysis (3D Gait Analysis and 6MWT) 

Twenty-two children completed the 6MWT. The 95% ICs for total distance covered and difference between sex and age are shown in Table 2. We found significant differences between ages (the distance travelled by the oldest children being significantly greater; *p* value = 0.02), while the results between sexes showed no significant differences.

Twenty-six children completed 3D gait analysis. The results showed that the average gait pattern of the sagittal and frontal planes was normal for all joints (pelvis, hip, knee, ankle, and foot).

The kinematic graphics in the transversal plane showed torsional alterations in the knee and hip joints. Both tibias were kept in external rotation during all gait cycles (oscillating between −18° and −8° on the right and −12° and −3° on the left), with a difference of 6° between the left (less external rotation) and right. In the hip, the graphics show the opposite of that in the tibia; the internal rotation of the right hip was greater than in the left hip (with a difference of 5° between right and left) (Figure 1).

The power graphs showed a decreased peak during the preswing and take-off phases in the sagittal plane of the ankle. In the knee, the preswing and take-off phases had positive values, whereas in the initial contact, these values were negative. On the hip, there was an absence of peak power during the initial swing phase (Figure 1).

The sagittal moment graphic showed a lowered peak during the end of terminal stance and preswing gait cycles in both ankles. In the moments of the sagittal plane of the right knee, it is important to highlight the increased moment at the beginning of the cycle (initial contact and loading response). In addition, during the end of pre-swing and initial swing, the moment was negative when it should have been positive. Considering knee kinematics, the initial contact and loading response were normal, but during the terminal stance and preswing, the moments stayed positive. The moment on the right hip in the sagittal plane remained stable during all gait cycles, without reaching any peak in the initial contact, final stance, preswing. or terminal swing (Figure 2).

The gait indicators GPS and GDI were 7.4 (±1.9) and 95.3 (±9.7), respectively, for the left side and 6.8 (±2.0) and 97.7 (±9.5), respectively, for the right side.

## 4. Discussion

This paper shows the results of a study of SS patients in whom little musculoskeletal involvement had been observed. Even though 30% of patients were found to have joint hyperlaxity, this finding did not seem to have repercussions on strength, functional tests, or gait alterations. This may have been due to several reasons, among which we would point to the low median age of the series [11] and because diagnosis of SS is perhaps more frequently performed by orofacial involvement and eye and hearing alterations. 

The finding that there were significant differences in the distance travelled in the 6MWT according to age may have been due to several factors. Perhaps one of the most important is that younger children have more difficulty walking for this reason alone, regardless of their basal capacity.

Chronic illness, primary bone disease, or poor nutrition in children and adolescents may lead to impaired skeletal health. Approximately 90% of adult bone mass is gained in the first two decades of life, and many experts think that optimizing peak bone mass and bone strength early in life and stabilizing it during young adulthood play a significant role in preventing osteoporosis and fractures later in life. There are several known determinants of skeletal mineralization and peak bone mass: genetics; race; gonadal status; sleep; and environmental factors such as nutrition and physical activity [21,26,27,28]. Some reports indicated that osteoporosis is common in SS, but this has not yet been systematically evaluated [11], and for these reasons, calcaneus QUS was measured in this study. In relation to bone mineral density, the SI is lower in children with SS than in European TD children (6 to 12 years (82.06 (12.43); >12 years (97.03 (16.09)) [26]. Similar findings were found in bone mineral density in children with juvenile idiopathic arthritis [29] and osteogenesis imperfecta [30], who presented an increased risk of fractures [30,31]. Moreover, the QUS calcaneal SI shoed moderate correlation (r = 0.69) with total body bone mineral density measured by dual-energy X-ray absorptiometry [21]. These findings probably corroborate the reduction in bone mineral density in SS, which could lead to an increased risk of fractures in childhood [32].

Children with SS showed slightly reduced values compared with TD children in GDI (TD 100 (SD 10) [25] and GPS (TD 5.3 (SD 1.4)) [24]. Moreover, the GPS difference was near to the minimal clinically important difference (MCID), which is 1.6°. We also found reduced ankle power during push-off and knee power at initial contact, while the rest of the joints exhibited results similar to those of TD children. The influence of reduced push-off force during gait has been reported in other studies. Huang T. W. et al. (2015) and Ong C. F. et al. (2019) applied gait simulator models to predict the adaptations to different type of deficits, finding an increase in work from other joints to maintain the gait velocity, while plantar flexor moments were minimally affected by plantar flexor weakness only [33,34]. Although children with SS have preserved plantar flexion strength, it appears that the propulsion strategy performed is not adequate. This means that the energy expenditure during gait will be higher and may be a factor for clinicians to consider when planning treatments.

Related to 6MWT results, we found significant differences in the distance traveled according to age due to several factors. Perhaps one of the most important is that younger children have more difficulty walking for this reason alone, regardless of their basal capacity.

The mean gait distance in 6MWT measured in meters in patients with SS at 95% IC (512 to 607) was lower than that reported in the literature for European TD children (619.8 (SD 58.3)) [24]. This difference was larger than the minimal clinically important difference, which ranges between 22.8  and 31 m [33,34,35,36]. Moreover, similar results were reported in children with juvenile idiopathic arthritis (478–602) [37], while this distance was lower in other pathologies that cause musculoskeletal disorders, such as hypophosphatasia (children: 350.4 (92.3); adolescents: 497.3 (98.8)) [33] or osteogenesis imperfecta type I (418 (175.0)) [38]. This finding suggests that children with SS have mild gait-related limitations, which could be a potential treatment target for SS patients. Therefore, this study provides new knowledge about the musculoskeletal characteristics of patients with SS. These findings may be useful to obtain general information on the functional status of these children, helping clinicians to better understand this pathology. However, a larger sample size and follow up are required to determine these bony misalignments as risk factors for joint pain in future studies.

### Limitations

Despite being a relatively large series of patients with this rare disease, this study is not without limitations. MMT testing has several disadvantages, including that testing muscle groups is time consuming, patients often experience fatigue during the testing and occasionally experience muscle pain that makes muscle testing unpleasant and stressful, and children frequently are not able to cooperate for the entire muscle group test, resulting in incomplete results or inconsistent strength evaluation. The selection of matched age and sex pairs could be a bias for comparison purposes due to the impact of the disease on children’s development. In addition, QUS is not the gold standard for bone mineral density measurement, so results should be interpreted with caution. Finally, the sample size was small, and no imputation was performed for missing data.

## 5. Conclusions

In this study, we described the musculoskeletal alterations in this sample of patients with SS. The impact of the disease on function is low, reporting less distance covered in the 6-min test and reduced bone mineral density compared with healthy children. Indeed, studies with follow-up periods are necessary to deepen our knowledge of what muscle-articular involvement of SS consists of in children, and the repercussions of this involvement on muscle strength and gait.

## Figures and Tables

**Figure 1 children-09-01895-f001:**
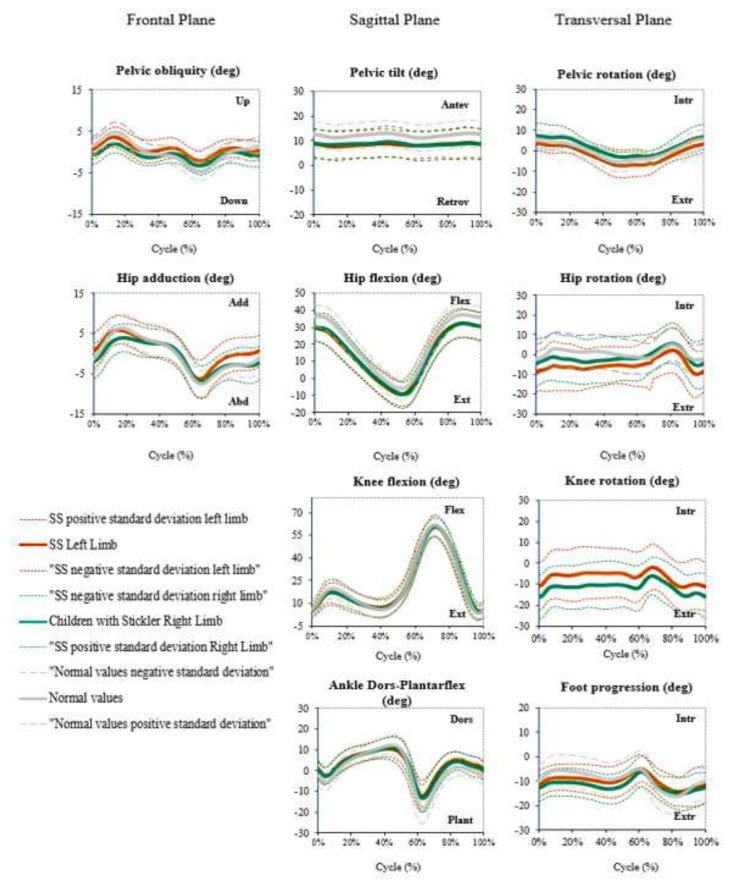
Kinematic 3D gait analysis of patients with SS compared with healthy children and adolescents. Abbreviations: Dors = dorsiflexion, SS = Stickler syndrome.

**Figure 2 children-09-01895-f002:**
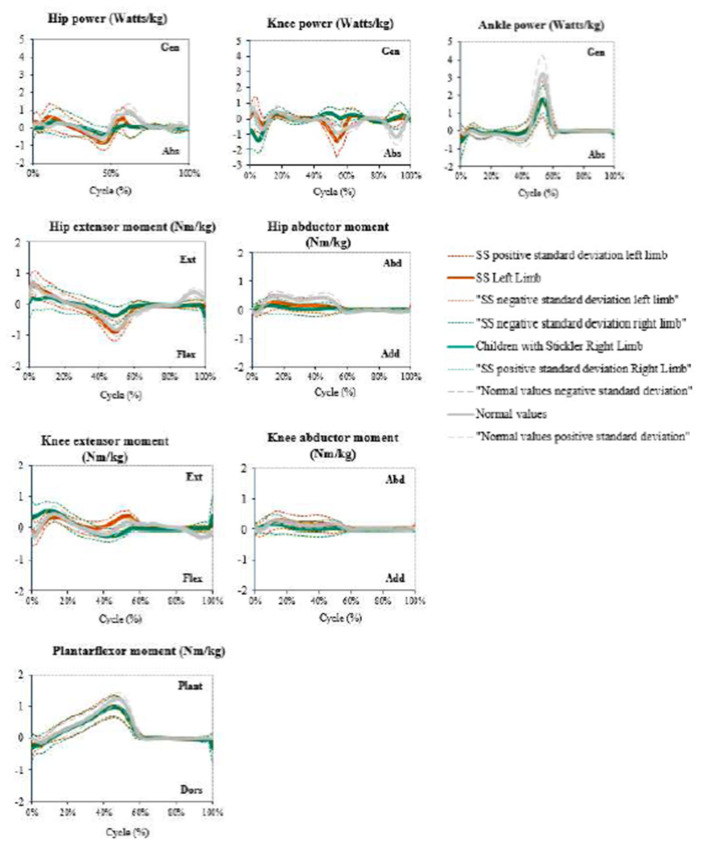
Kinetic analysis of patients with SS compared with healthy children and adolescents. Abbreviations: SS = Stickler syndrome.

**Table 1 children-09-01895-t001:** Stickler Syndrome’s features, functional tests.

Musculoskeletal SS’s Features (n = 23)			
Variables	n	Value n (%)	95% confidence interval
Osteoarthritis	23	4 (17)	0.6 to 34.1
Ligament hyperlaxity syndrome	23	7 (30)	10.1 to 50.7
Marfanoid habit	23	4 (17)	0.6 to 34.1
Spinal dysplasia	23	2 (9)	−3.8 to 21.1
Muscular atrophy	23	4 (17)	0.6 to 34.1
**Functional Tests (n = 25)**			
Variables	n	Value (%)	
Duncan–Ely test	25	3 (12)	
Galeazzi or Allis test	25	2 (8)	
Thomas test	25	4 (16)	
Silfverskiold test	25	1 (4)	
Alteration in any test	25	7 (28)	

Abbreviations: SS = Stickler syndrome.

**Table 2 children-09-01895-t002:** Bone and gait characteristics of children with Stickler syndrome.

Bone and Gait Characteristics			
Outcomes	Sample (n)	95% CI	Difference
Calcaneus SI	Total (26)	71.8 to 87.4	-
	Male (9)	61.7 to 85.1	−24.9 to 5.9
	Female (17)	72.8 to 93.0	
	Under 10 years (13)	60.2 to 81.6	**−31.7 to −3.1** *
	Over 10 years (13)	78.8 to 97.8	
6MWT distance (m)	Total (22)	512.7 to 607.5	-
	Male (7)	445.7 to 646.7	−134.3 to 93.7
	Female (15)	512.8 to 620.2	
	Under 10 years (11)	453.1 to 571.4	**−183.329 to −8.071** *
	Over 10 years (11)	543.3 to 672.6	
GPS	Left side (26)	6.7 to 8.1	-
	Right side (26)	6.0 to 7.6	
GDI	Left side (26)	91.6 to 99.0	-
	Right side (26)	94.0 to 101.4	

Bold values denote statistical significance: (*) *p* < 0.05 level. Abbreviations: CI = confidence interval; GDI = gait deviation index; GPS = gait profile score; MWT = meter walking test; SI = stiffness index; y = years old.

## Data Availability

Not applicable.

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
