# Peer review of "Musculoskeletal and Gait Characteristics in Patients with Stickler Syndrome: A Cross-Sectional Study"

_children, 2022, doi:10.3390/children9121895_

Round 1

Reviewer 1 Report

Stickler syndrome (SS) is a dominantly inherited connective tissue disorder of the fibrillary collagen with very variable clinical manifestation, including osteoarthritis and osteopenia. Heterogenous mutations of COL2A1, COL11A1, COL11A21, and COL9A1 procollagen genes lead to various degrees of abnormal synthesis of collagen types II, XI, and IX. Two opthalmological phenotypes, membranous and bedded, are used to classify SS into two different types. Clinical manifestations are highly variable and distributed in four large groups including 1) craniofacial findings; 2) eye alterations; 3) hearing impairment; 4) musculoskeletal features, especially early-onset arthropathy, short stature, and mild spondyloepiphyseal dysplasia. The presence of musculoskeletal disorders in children may affect gait maturity or strength difficulties. The aim of this study is to analyze the musculoskeletal characteristics, gait kinematics and kinetics in SS patients. 

A cross-sectional study design include X patients, with appropriate approval obtained by the ethics committee. Patient eligibility was described. Musculoskeletal characteristics were described, and muscular strength and functional tests were performed. Counts of patients presenting with pathological results of these tests were collected. Calcaneus quantitative ultrasound was performed as a quick, cost efficient, and radiation free method to evaluate bone stiffness that indicates density, structure, and composition of the bone. 3D gait analysis was performed using eight optoelectronic cameras and two dynamometric platforms. These data were compared with a sample of healthy children and adolescents (n=25) with the aim to describe gait alterations in children and adolescents of similar ages. Statistical analysis was performed using non-parametric tests due to small sample size.

Table 1 presents a summary of the main musculoskeletal features of these patients with SS. The most common alteration was ligament hyperlaxitude syndrome (30%) followed by osteoarthritis, marfanoid habit, and muscular atrophy (17% each). Table 2 presents results of the 5 minute walking test, and includes differences between men and women and individuals under 10 years and over 10 years of age. There were no differences by sex, but the oldest children traveled significantly greater distances than younger children. Table 3 provided a summary of QUS results in a similar manner. Values were significantly different between younger and older children. 3D gait analysis revealed that the average gait patter of the sagittal and frontal planes was normal for all joints (pelvis, hip, knee, ankle, and foot). Kinematic graphics in transversal plane showed torsional alterations in knee and hip joints. Both tibias were kept in external rotation during all gait cycle, with a difference of 6 degrees between left and right. In the hip, the graphics show the opposite. Results were graphically shown in Figure 1.

Results indicated that the patients in this study had little musculoskeletal involvement. While 30% of patients suffered joint hyperlaxitude, no effect was seen on strength, functional tests or gait alterations. Significant differences were reported in 6MWT of the age groups in patients with SS. Results indicated a lower 6MWT distance than reported in the literature for healthy European children. SI, as determined by QUS, was also lower in patients with SS when compared to reported results in the literature for healthy European children. Limitations of the study included small patient number, and inherent disadvantages of MMT and QUS tests.

Suggested changes:

1. Sample size should be included in the paragraph describing patients in the Materials and Methods section. Sample size is not mentioned until results.

2. What type/brand of ultrasound was used for QUS? Are there any scanning parameters which should be added to the methodology?

3. What software was used to conduct the statistical analysis?

4. Results – paragraph one – missing period at end.

5. Results – this is the first description of the sample. I feel this data should be moved to materials and methods under a “subjects” or other heading to describe the population being discussed.

6. Results – it appears as though line 149 is meant to be a heading similar to line 162. This needs to be corrected and

7. Figure 1 – please consider all types of colorblindness when choosing graphic colors. Some individuals may not be able to differentiate between the red and green lines used for “SS left limb” and “Children with Stickler Right Limb”. Additionally, the naming of the lines should be standardized. Quotation marks within the legend do not appear necessary. Consider reviewing color blindness guidelines provided by MDPI or Color University Design (https://jfly.uni-koeln.de/color/) to assist in finding appropriate graphics colors. Not all abbreviations are described. If appropriate to the journal, consider adding additional identifiers for individual tests A, B, C, etc.

8. Figure 2 suffers from the same issues as Figure 1 regarding color, standardization of names, clarity of axes labels, and legend. Not all abbreviations are described.

9. Should line 194 refer to Figure 2? There does not appear to be a Figure 3 included.

10. Should ligament hyperlaxitude syndrome be ligament hyperlaxity syndrome?

11. Please review the literature again for more information that could be useful in the introduction and discussion. 

12. As the control for multiple metrics in this manuscript are those found in literature, a more thorough description of that data should be included and/or discussed in the discussion.

Author Response

We are thankful to the referees for deeply analyze the manuscript. We had seen those contributions as a very helpful points to develop a better analysis of this work.

Reviewer 1.

1.- Sample size should be included in the paragraph describing patients in the Materials and Methods section. Sample size is not mentioned until results.

We added information about the sample size in the M&M  section.

  1. What type/brand of ultrasound was used for QUS?

The information related to the Brand was adeed in M&M section, “QUS was measured using a Achiles EXP II system (GE Heathcare).”

Are there any scanning parameters which should be added to the methodology?

The scanning procedure is not provided by the commercial brand. A standardized protocol for data adquisition was using following the technical instructions.

  1. What software was used to conduct the statistical analysis?

Information regarding the software was included in the M & M section. “IBM Corp. Released 2020. IBM SPSS Statistics for Windows, Version 27.0. Armonk, NY: IBM Corp”

  1. Results – paragraph one – missing period at end.

Text is completed with over 5 in manual muscle testing.

  1. Results – this is the first description of the sample. I feel this data should be moved to materials and methods under a “subjects” or other heading to describe the population being discussed.

We use the description of the sample as one of the first results reported in this study.

  1. Results – it appears as though line 149 is meant to be a heading similar to line 162. This needs to be corrected and
  2. Figure 1 – please consider all types of colorblindness when choosing graphic colors. Some individuals may not be able to differentiate between the red and green lines used for “SS left limb” and “Children with Stickler Right Limb”. Additionally, the naming of the lines should be standardized. Quotation marks within the legend do not appear necessary. Consider reviewing color blindness guidelines provided by MDPI or Color University Design (https://jfly.uni-koeln.de/color/) to assist in finding appropriate graphics colors. Not all abbreviations are described. If appropriate to the journal, consider adding additional identifiers for individual tests A, B, C, etc.
  3. Figure 2 suffers from the same issues as Figure 1 regarding color, standardization of names, clarity of axes labels, and legend. Not all abbreviations are described.

Dear reviewer, we are working on a new desing of the figure for adaptation to the guidelines.

  1. Should line 194 refer to Figure 2? There does not appear to be a Figure 3 included.

We correct the mistake.

  1. Should ligament hyperlaxitude syndrome be ligament hyperlaxity syndrome?

We correct the mistake.

  1. Please review the literature again for more information that could be useful in the introduction and discussion. 

We completed the discussion section

  1. As the control for multiple metrics in this manuscript are those found in literature, a more thorough description of that data should be included and/or discussed in the discussion.

We completed the discussion section.

Reviewer 2 Report

This study investigated the effect of the Stickler syndrome on the strenght and gait of children between 4 to 18 years. The study is clearily described and the statistics support the results. However, I have some questions and comments that I think it may improve the quality and the readability of the study.

Authors found a negative result: the SS did not affected the gait and strength of patients. Therefore,I suggest to better explicit why they were expecting an effect, e.g. reporting literature that may lead to the hypothesis that SS could influence strenght and gait.

Are there some literature or are you planning to perform the same experiment on adult patients to investigate whether alterations in gait occurred later?

How can you assess the performance of SS patients? Are you expecting that recruiting healthy children, age and gender matching, as controls and reference may influence your results?

Since puberty leads to phyiological changes, I suggest to perform your analysis on the two separate groups of pre-puberal and post-puberty childrens to detect whether effects could be identified in only one of these subpopulations.

Author Response

We are thankful to the referees for deeply analyze the manuscript. We had seen those contributions as a very helpful points to develop a better analysis of this work.

Authors found a negative result: the SS did not affected the gait and strength of patients. Therefore,I suggest to better explicit why they were expecting an effect, e.g. reporting literature that may lead to the hypothesis that SS could influence strenght and gait.

The initial hypotheses of the research project was related to the clinical perspective of the authors, according to similar results founded in other populations (Ostogenesis imperfecta or hypophosphtasia).

Are there some literature or are you planning to perform the same experiment on adult patients to investigate whether alterations in gait occurred later?

At this moment we´re only attending to pediatric populations, anyway we are in contact with colleagues that could develop research projects in this direction in the near future. Thanks for the idea.

How can you assess the performance of SS patients? Are you expecting that recruiting healthy children, age and gender matching, as controls and reference may influence your results?

This is a descriptive study, we add this as a potential limitation of the study.

The selection of matched age and gender pair could be a bias for comparison purposed due to the impact of the disease in children development.

Since puberty leads to phyiological changes, I suggest to perform your analysis on the two separate groups of pre-puberal and post-puberty childrens to detect whether effects could be identified in only one of these subpopulations.

We are agree with your view, but due to the small sample size of each population a separate analysis could lead to confusing conclusions. We review the data and perform a different analysis.

Round 2

Reviewer 1 Report

Dear Authors,

The overall improvements to the manuscript are quite good. However, I must stress that the red and green colors chosen for lines are very difficult to distinguish if one suffers from  dueteranopia-type color blindness. I have attached an image showing what the figures looks like to someone with this type of color blindness to be as clear as possible. In my previous review, I gave suggestions for finding colors that will work together. If one cannot distinguish what lines are which using greyscale, either, then the figure needs work. Please update the colors for the figures so that all readers may be able to comprehend them. If you have Photoshop available to you, it is possible to review the images as one with colorblindness by selecting View from the main menu, then "Proof Setup" and choosing "Color Blindness - Deuteranopia-type" or "Color Blindness - Protanopia-type". There are also options online for reviewing images.

Author Response

Dear reviewer.

Here attacched the new version of the figures.

Thanks for your improvements.

Sergio Lerma
